EMBO
Molecular Medicine

# USP9X stabilizes XIAP to regulate mitotic cell death and chemoresistance in aggressive B-cell lymphoma

Katharina Engel[1], Martina Rudelius[2], Jolanta Slawska[1], Laura Jacobs[1], Behnaz Ahangarian Abhari[3], Bettina Altmann[4], Julia Kurutz[1], Abirami Rathakrishnan[1], Vanesa Fernández-Sáiz[1,5], Andrä Brunner[1], Bianca-Sabrina Targosz[1], Felicia Loewecke[1], Christian Johannes Gloeckner[6,7], Marius Ueffing[6], Simone Fulda[3,5], Michael Pfreundschuh[8], Lorenz Trümper[9], Wolfram Klapper[10], Ulrich Keller[1,5], Philipp J Jost[1], Andreas Rosenwald[2], Christian Peschel[1,5] & Florian Bassermann[1,5,*]

## Abstract

The mitotic spindle assembly checkpoint (SAC) maintains genome stability and marks an important target for antineoplastic therapies. However, it has remained unclear how cells execute cell fate decisions under conditions of SAC-induced mitotic arrest. Here, we identify USP9X as the mitotic deubiquitinase of the X-linked inhibitor of apoptosis protein (XIAP) and demonstrate that deubiquitylation and stabilization of XIAP by USP9X lead to increased resistance toward mitotic spindle poisons. We find that primary human aggressive B-cell lymphoma samples exhibit high USP9X expression that correlate with XIAP overexpression. We show that high USP9X/XIAP expression is associated with shorter event-free survival in patients treated with spindle poison-containing chemotherapy. Accordingly, aggressive B-cell lymphoma lines with USP9X and associated XIAP overexpression exhibit increased chemoresistance, reversed by specific inhibition of either USP9X or XIAP. Moreover, knockdown of USP9X or XIAP significantly delays lymphoma development and increases sensitivity to spindle poisons in a murine Eμ-Myc lymphoma model. Together, we specify the USP9X–XIAP axis as a regulator of the mitotic cell fate decision and propose that USP9X and XIAP are potential prognostic biomarkers and therapeutic targets in aggressive B-cell lymphoma.

**Keywords** B-cell lymphoma; mitosis; ubiquitin; USP9X; XIAP
**Subject Categories** Cancer; Haematology

## Introduction

The spindle assembly checkpoint (SAC) is an important target in cancer therapy, and microtubule poisons such as taxanes and vinca alkaloids are widely used as highly effective chemotherapy agents (Manchado *et al*, 2012). The failure of tumor cells to initiate apoptosis during the active checkpoint signal is a major factor that limits the efficacy of antimitotic drugs (Janssen & Medema, 2011). Therefore, the identification of factors that regulate mitotic survival is of imminent relevance, both in generating a mechanistic understanding of the SAC and for the identification of novel target structures in cancer therapy.

Pathways of the ubiquitin proteasome system are critically involved in SAC signaling, mediating both proteolysis-dependent and proteolysis-independent mechanisms (Lara-Gonzalez *et al*, 2012). Indeed, the target of the SAC is CDC20, which activates the anaphase-promoting complex/cyclosome (APC/C) that triggers anaphase and mitotic exit by ubiquitylating securin and cyclin B1 (Lara-Gonzalez *et al*, 2012). Other ubiquitin ligases, particularly members of the cullin ring ligase (CRL) family, regulate different aspects of mitotic spindle formation, mitotic survival, and chromosome alignment, respectively (Bassermann *et al*, 2014). By contrast, the role of deubiquitylases (DUBs) in mitotic control has remained more elusive.

Inhibitor of apoptosis proteins (IAP) are key negative regulators of cell death. Specific to the IAP member X-linked inhibitor of apoptosis protein (XIAP) is the ability to directly bind and inhibit activated caspases 3, 7, and 9 via its N-terminal baculoviral IAP repeat domains (BIR) (Deveraux *et al*, 1997; Chai *et al*, 2001; Riedl *et al*, 2001; Shiozaki *et al*, 2003). Notably, XIAP exerts ubiquitin ligase

1 Department of Medicine III, Klinikum Rechts der Isar, Technische Universität München, München, Germany
2 Institute of Pathology and Comprehensive Cancer Center Mainfranken, Universität Würzburg, Würzburg, Germany
3 Institut für Experimentelle Tumorforschung in der Pädiatrie, Goethe-Universität Frankfurt, Frankfurt am Main, Germany
4 Institute for Medical Informatics, Statistics and Epidemiology, Universität Leipzig, Leipzig, Germany
5 German Cancer Consortium (DKTK), German Cancer Research Center (DKFZ), Heidelberg, Germany
6 Eberhard-Karls-Universität Tübingen, Institute for Ophthalmic Research, Medical Proteome Center, Tübingen, Germany
7 German Center for Neurodegenerative Diseases (DZNE), Tübingen, Germany
8 Department of Medicine I, Saarland University Medical School, Homburg (Saar), Germany
9 Department of Hematology and Oncology, Georg-August-Universität Göttingen, Göttingen, Germany
10 Institute of Pathology, Haematopathology Section and Lymph Node Registry, Universitätsklinikum Schleswig-Holstein, Kiel, Germany
*Corresponding author. Tel: +49 89 4140 5038; E-mail: florian.bassermann@tum.de

activity and promotes ubiquitylation of substrates such as RIPK2, but also regulates its own abundance via autoubiquitylation, an important mechanism for maintaining XIAP homeostasis (Vaux & Silke, 2005) (Damgaard et al, 2012). Increased expression of XIAP is found in different cancers, including B-cell lymphomas; however, the functional relevance has not been fully elucidated (Akyurek et al, 2006).

Diffuse large B-cell lymphoma (DLBCL) is the most common aggressive B-cell lymphoma in adults (Shaffer et al, 2012). The majority of patients achieve a remission of their disease after first-line immunochemotherapy, which typically includes vinca alkaloids and checkpoint inducing agents such as anthracyclines, but 10% of patients are upfront refractory. Notably, about one-third of initially treatment-responsive tumors eventually recur or become refractory to conventional therapy (Martelli et al, 2013), indicating the urgent need to develop more specific targeted therapies. While a number of risk factors for aggressive disease as well as adverse genetic profiles and histologic categories have been defined (Ziepert et al, 2010; Shaffer et al, 2012; Pasqualucci, 2013), the molecular mechanisms of treatment resistance and escape from mitotic cell death remain obscure.

In this study, we set out to investigate the role of DUBs in the regulation of mitotic cell fate decisions. These studies revealed a mitosis-specific function for the ubiquitin-specific protease 9X (USP9X), which we show to deubiquitylate and stabilize XIAP in order to promote mitotic survival. We demonstrate that loss of this anti-apoptotic pathway sensitizes cells to spindle poison-induced mitotic cell death independent of their MCL1 status and show that overexpression of USP9X and concomitant high expression of XIAP determine adverse prognosis and treatment resistance in human aggressive B-cell lymphomas.

## Results and Discussion

To begin assessing the role of deubiquitylation in mitotic control, we analyzed cell cycle profiles of different USP family DUBs. This approach identified a significant enrichment of USP9X in mitotic cells (Fig 1A). USP9X is involved in fundamental processes such as pre-implantation development and is considered to particularly enhance cell survival by stabilizing different pro-survival substrates such as β-catenin and the pro-survival BCL2 family member MCL1 (Taya et al, 1999; Murray et al, 2004; Sacco et al, 2010; Schwickart et al, 2010; Vucic et al, 2011). While these pro-survival activities have not been shown to occur in a cell cycle-dependent manner, USP9X demonstrates mitotic activity as evidenced by its role in regulating chromosome alignment and segregation by means of targeting the inhibitor of apoptosis protein (IAP) family member survivin and Aurora B to centromeres (Vong et al, 2005). However, this mechanism coordinates proper chromosome division rather than cell survival. We therefore reasoned that USP9X might exert mitotic pro-survival activity by stabilizing other IAP family members than survivin. To investigate this possibility, we analyzed the stability of different IAP proteins under the condition of RNAi-mediated USP9X knockdown. We found that XIAP was the only candidate that displayed significant loss of mitotic expression in USP9X-silenced cells that was restored upon treatment with the proteasome inhibitor MG132, thus indicating a role for USP9X in

proteasome-dependent mitotic stabilization of XIAP (Fig 1B and Appendix Fig S1A and B). In further support of this notion, we found that both XIAP and USP9X display parallel accumulation in mitosis (Fig 1A).

To further substantiate a role for USP9X in regulating XIAP in mitosis, we tested direct interaction of USP9X and XIAP. Indeed, co-immunoprecipitation studies under semi-endogenous conditions identified preferential interaction of XIAP with USP9X in mitosis (Fig 1C and Appendix Fig S1C). Likewise, immunofluorescence studies revealed co-localization of USP9X and XIAP in either prometaphase or metaphase cells (Appendix Fig S1D). Moreover, the interaction of USP9X and XIAP increased upon prolonged exposure to taxol, which leads to a gradual mitotic enrichment of cells (Appendix Fig S1E). Direct interaction of USP9X with XIAP was further verified in vitro using purified proteins (Fig EV1A). Notably, XIAP specifically interacted with the USP9X fragment containing the active cystein protease site (Fig EV1A).

Mapping studies using different deletion mutants narrowed the USP9X binding motif to the BIR2 and BIR3 domains of XIAP (aa152–323) (Fig EV1B). This sequence contains a glycine residue at position 188, whose germline mutation marks a causative molecular aberration of the X-linked lymphoproliferative syndrome type 2 (XLP-2), which features low or instable expression of XIAP and premature apoptosis of lymphocytes in response to different stimuli (Rigaud et al, 2006). Two respective mutations of XIAP at G188 have been described, G188E and G188R, which diminish XIAP expression levels in patient B cells (Marsh et al, 2009; Gifford et al, 2014). Indeed, we found that both mutations abolish binding of XIAP to USP9X, and both XIAP mutants demonstrate reduced stability in mitosis (Figs 1D and EV1C). Moreover, SMAC mimetics, which are known to interfere with protein binding of XIAP via its BIR2 domain (Krieg et al, 2009; Damgaard et al, 2013), disrupted the interaction of USP9X to XIAP (Fig EV1D). Together, these data suggest a role of Gly188 within the BIR2 domain as the binding site of XIAP for USP9X.

We next investigated whether mitotic stability of XIAP directly relies on the deubiquitylating activity of USP9X. To this end, we performed deubiquitylation assays using mitotic cells under conditions of both USP9X knockdown and forced USP9X expression. Indeed, ubiquitylation of XIAP was substantially increased upon silencing or chemical inhibition of USP9X (Figs 1E and EV2A) in mitotic cells, while forced expression of USP9X attenuated XIAP ubiquitylation (Fig EV2A). In line with this, we found the overall deubiquitylation activity of USP9X to be elevated in mitosis (Fig EV2B). Notably, staining with linkage-specific ubiquitin antibodies revealed that USP9X removes K48-linked ubiquitin chains from XIAP (Fig EV2C). Moreover, we found that ubiquitylation of the XIAP$^{G188R}$ mutant is substantially increased in mitotic cells as compared to WT XIAP and that mitotic ubiquitylation of XIAP$^{G188E}$ remained unaffected upon USP9X overexpression (Fig EV2D and E). These findings support the notion that the reduced stability of these mutants may result from their inability to bind USP9X with the consequence of increased ubiquitylation and degradation, and may have implications in the pathophysiology of the XLP-2 syndrome. In a complimentary approach, we found that a catalytically inactive USP9X mutant (USP9X$^{C1556S}$) was unable to confer stability to XIAP in mitotic cells (Fig 1F). Likewise, addition of the USP9X inhibitor WP1130 destabilized XIAP in mitotic cells (Fig 1G).

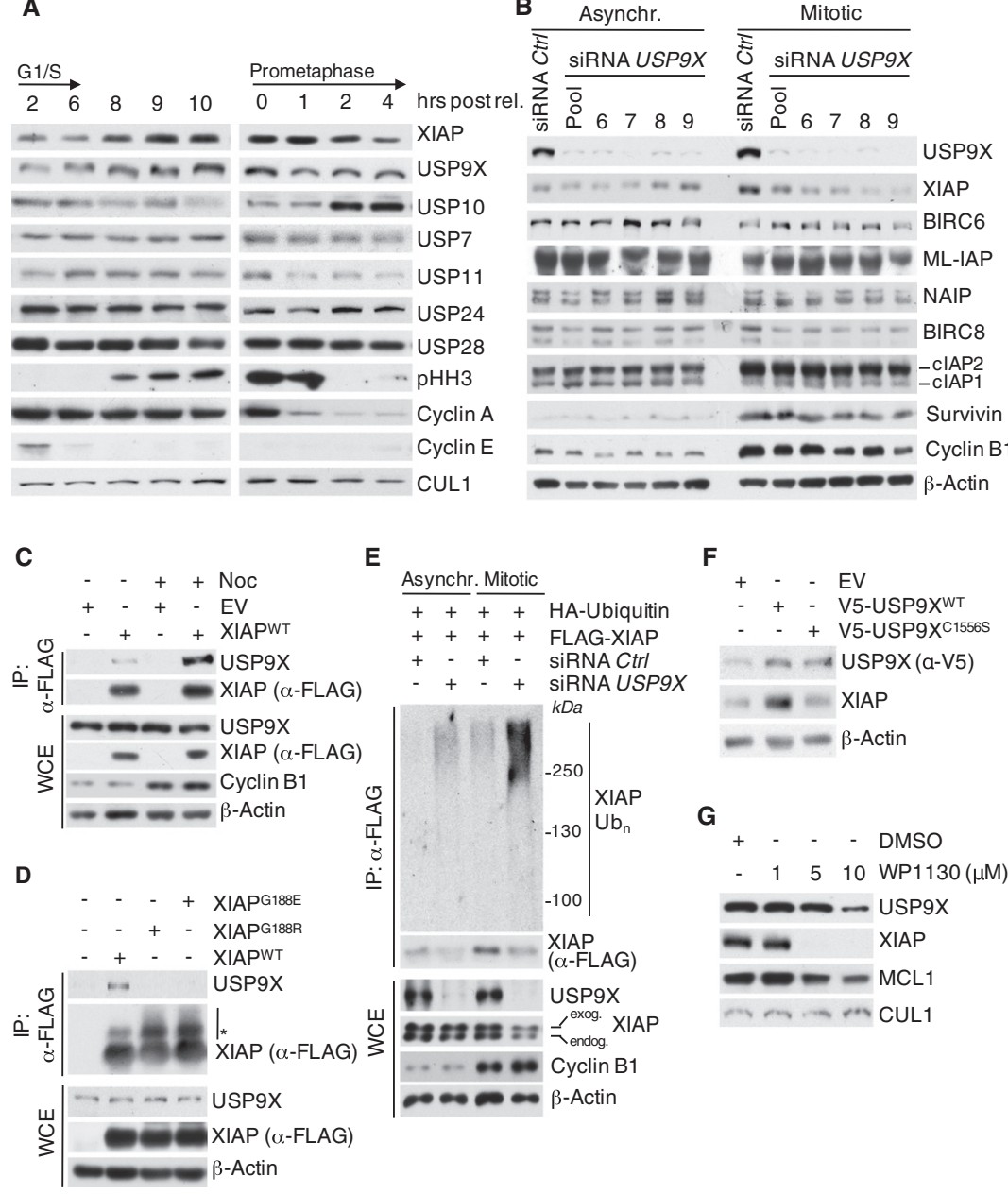

**Figure 1.  USP9X binds, deubiquitylates, and stabilizes XIAP in mitosis.**

A   Immunoblot analysis of HeLa cells using the indicated antibodies that were synchronized in G1/S phase using a double thymidine block or in mitosis using sequential thymidine and nocodazole treatment and collected at the given time points.

B   Immunoblot analysis of HeLa cells that were transfected with siRNA oligonucleotides as indicated and left untreated or synchronized in mitosis using sequential thymidine and nocodazole treatment.

C   Co-immunoprecipitation of XIAP and endogenous USP9X in HEK 293T cells that were transfected with FLAG-tagged XIAP or empty vector (EV) and synchronized in mitosis using nocodazole or left untreated.

D   Co-immunoprecipitation of XIAP and endogenous USP9X in HEK 293T cells that were treated as in (C), but with additional transfection of the FLAG-tagged XIAP single amino acid mutants G188R and G188E. Only mitotic samples are shown. The asterisk denotes ubiquitylated forms of XIAP.

E   *In vivo* ubiquitylation of XIAP in HeLa cells that were infected with the indicated expression constructs carrying FLAG-tagged XIAP and transfected with siRNA oligonucleotides as specified. Cells were synchronized in mitosis using sequential thymidine/nocodazole treatment, as indicated. Subsequent to treatment with MG132, whole-cell extracts (WCE) were prepared and ubiquitylated XIAP was isolated by anti-FLAG immunoprecipitation (IP) under denaturing conditions.

F   Immunoblot analysis of NIH 3T3 cells that were transfected with expression constructs for USP9X[WT] or the catalytically inactive mutant USP9X[C1556S]. The band in the EV control lane of the anti-V5 panel marks an unspecific band produced by the antibody.

G   Immunoblot analysis of HeLa cells using antibodies to the indicated endogenous proteins that were synchronized in mitosis using thymidine/nocodazole and treated with DMSO or the USP9X inhibitor WP1130 as indicated.

Source data are available online for this figure.

We reasoned that USP9X deubiquitylates XIAP to regulate mitotic survival. To investigate this possibility, we silenced USP9X expression and analyzed cell death under conditions of prolonged taxol-induced mitotic arrest. Indeed, suppression of the USP9X–XIAP axis increased mitotic cell death in wild-type MEFs as evidenced by an increase of the sub-G1 fraction and a decrease of the mitotic cell population as well as by an increase of cleaved caspase-3 and loss of cyclin B1 expression (Fig 2A and B). Importantly, *Usp9X* knockdown did not induce specific loss of the mitotic compartment in *Xiap*$^{-/-}$ cells, indicating that USP9X promotes mitotic survival directly via stabilization of XIAP (Fig 2B). Of further notice, *Xiap*$^{-/-}$, but not *cIap2*$^{-/-}$, cells demonstrate an increased overall sensitivity toward taxol exposure in the control setting, thus further emphasizing a role for XIAP in preventing mitotic apoptosis (Fig 2A and B). A previous study suggested MCL1 as a cell cycle-independent substrate of USP9X (Schwickart *et al*, 2010). In addition, MCL1 has been shown to be ubiquitylated and degraded via APC/C$^{Cdc20}$ and SCF$^{Fbxw7}$ upon onset of mitosis (Harley *et al*, 2010; Inuzuka *et al*, 2011; Wertz *et al*, 2011). We therefore investigated the effect of *Usp9X* knockdown in *Mcl1*$^{-/-}$ MEFs to rule out the possibility that USP9X mediates its mitosis-specific effects by stabilizing MCL1. Indeed, knockdown of *Usp9X* increased mitotic cell death in *Mcl1*$^{-/-}$ cells to a similar extent as in WT MEFs, thus indicating independence of MCL1 (Fig 2A and B). The elevated baseline apoptosis observed in *Mcl1*$^{-/-}$ cells is likely attributable to elevated spontaneous apoptosis and higher sensitivity to infection procedures, which have been reported for these cells (Opferman *et al*, 2005). In line with these observations, neither USP9X overexpression nor *USP9X* knockdown affected expression levels of MCL1 in mitotically arrested cells (Figs 1G and 2A). Finally, we investigated the impact of forced USP9X and XIAP expression as well as *XIAP* silencing on mitotic survival. In accordance with the results above, overexpression of either USP9X or XIAP protected cells from apoptosis, as visualized by a decrease of caspase-3 cleavage (Fig 2C and D) and the reduction of the apoptotic index (Fig 2E). By contrast, this effect was not observed upon expression of the USP9X binding-deficient XIAP$^{G188R}$ and XIAP$^{G188E}$ mutants, presumably because of their decreased mitotic stability (Fig EV1C). Likewise, loss of XIAP promoted cell death upon prolonged taxol-induced mitotic arrest (Fig 2F).

Together, the above results demonstrate that USP9X confers mitotic stability to XIAP during the activated SAC by means of interaction with XIAP at glycine residue 188 and subsequent specific deubiquitylation, thereby mediating mitotic survival independent of MCL1. Of notice, unchallenged XIAP-deficient mice are viable and do not show any substantial histopathological abnormalities (Harlin *et al*, 2001; Olayioye *et al*, 2005). Our results suggest that XIAP may be of particular relevance in a cellular environment that challenges mitotic survival.

The evasion of apoptosis is particularly pertinent to B-cell lymphomagenesis, where in the cells of origin, DNA double-strand breaks are a physiological process during the formation and revision of antigen receptors, requiring efficient means to eliminate cells with persistent DNA damage. In addition, agents like vinca alkaloids that target the mitotic checkpoint are key components of current first-line lymphoma treatment protocols (Shaffer *et al*, 2012; Wilson, 2013). We therefore investigated whether the USP9X–XIAP circuit is involved in the pathophysiology and therapy resistance of B-cell

lymphomas. This effort was further fostered by different lymphoma aCGH and NGS studies that suggest overexpression and copy number gains of USP9X in human diffuse large B-cell lymphoma (DLBCL) (Bea *et al*, 2005; Morin *et al*, 2013).

First, we investigated the role of USP9X-mediated stabilization of XIAP in the context of B-cell lymphoma maintenance *in vivo*. For these studies, we chose a syngeneic approach using primary Eµ-Myc B-cell lymphoma-derived cells. The Eµ-Myc mouse model is well established to study alterations in apoptotic pathways that drive B-cell lymphoma development and maintenance (Adams *et al*, 1985; Vaux *et al*, 1988; Eischen *et al*, 1999). Given the short latency of this approach, we chose to silence either *Usp9X* or *Xiap* (Fig 3A), which would be predicted to delay disease onset. Indeed, lymphoma onset was substantially delayed in mice receiving either *Usp9X*- or *Xiap*-silenced cells (Fig 3B and C), in line with an *in vitro* analysis of the respective cells (Appendix Fig S2A). At necropsy, lesions were confirmed to be B-cell lymphoma, and increased apoptosis as well as reduced infiltration of lymph nodes and bone marrow was present in both *Usp9X*- and *Xiap*-silenced specimens (Fig 3D and E, and Appendix Fig S2B and C). In a complimentary approach using PET analysis, we found significantly reduced tumor burden of mice receiving *Usp9X*- or *Xiap*-silenced cells 2 weeks after lymphoma cell injection (Fig 3F). Of notice, simultaneous knockdown of *Usp9X* and *Xiap* did not increase survival as compared to the single *Xiap* knockdown, indicating that USP9X functions via XIAP to promote lymphoma survival and growth in this model (Appendix Fig S2D and E). Importantly, response to vincristine treatment was substantially increased in mice receiving *Usp9X* or *Xiap* knockdown cells (Fig 3G and H, and Appendix Fig S2F). These data suggest that USP9X-mediated stabilization of XIAP promotes growth, survival, and resistance to spindle poisons in B-cell lymphoma.

We next investigated the role of the USP9X–XIAP axis in human DLBCL cell lines.

To this end, we first specified human DLBCL cell lines with either low (SuDHL 4, SuDHL 6, Oci-Ly 3) or high (RIVA, HT, Oci-Ly 10) USP9X expression and found associated high expression of XIAP (Fig 4A). Notably, RIVA, HT, and Oci-Ly 10 cells demonstrate increased resistance toward taxol treatment (Fig 4B). To investigate a direct functional involvement of USP9X, we silenced *USP9X* in these cells and treated with either taxol or doxorubicin, which typically induces G2/M arrest in tumor cells via activation of the DNA damage response. Indeed, USP9X knockdown resolved resistance to taxol or doxorubicin in USP9X/XIAP high-expressing cells (HT, RIVA) as demonstrated by caspase-3 cleavage, while the USP9X/ XIAP low-expressing cell lines SuDHL 4 and SuDHL 6 were not significantly affected (Figs 4C and EV3A). Similar results were observed in a complementary approach using the USP9X inhibitor WP1130 in SuDHL 6, Oci-Ly 3, HT, and Oci-Ly 10 cells treated with taxol (Kapuria *et al*, 2010; Wang *et al*, 2014; Figs 4D and EV3B).

X-linked inhibitor of apoptosis protein is naturally inhibited by SMAC (second mitochondria-derived activator of caspases)/DIABLO (direct inhibitor of apoptosis-binding protein with low isoelectric point) binding, which results in the release and activation of caspase-9 (Du *et al*, 2000). Small-molecule inhibitors mimicking this effect, that is, SMAC mimetics, have shown potent apoptosis-inducing activity in cell culture, animal models, and early clinical trials (Fulda & Vucic, 2012). We therefore treated DLBCL cells with the SMAC mimetic BV6 and/or taxol. Indeed, we observed significant

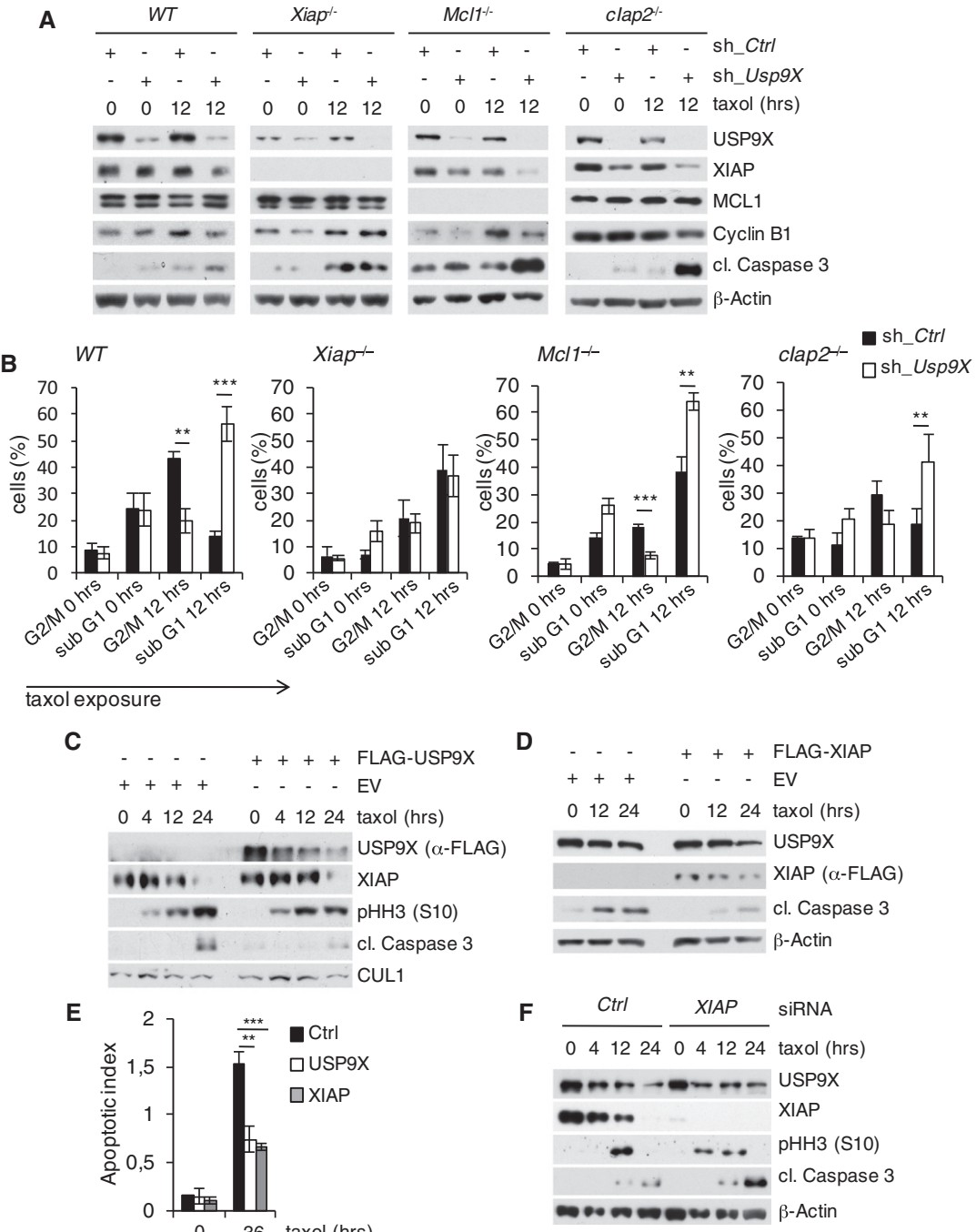

**Figure 2.  USP9X stabilizes XIAP to antagonize mitotic cell death independent of MCL1.**

A   Immunoblot analysis of WT, $Xiap^{-/-}$, $Mcl1^{-/-}$, or $cIap2^{-/-}$ mouse embryonic fibroblasts that were lentivirally infected with shRNA constructs directed against a non-relevant mRNA (Ctrl) or against $Usp9X$ mRNA and treated with taxol as specified.

B   Two-dimensional cell cycle analysis (BrdU/PI) of cells described in (A). Sub-G1 and G2/M fractions of cells were quantified and averaged with two additional, independent experiments ($n = 3$, $\pm$ SD; WT: **$P = 0.0016$; ***$P = 0.0004$; $Mcl1^{-/-}$: ***$P = 0.0003$; **$P = 0.0017$; $cIap2^{-/-}$: **$P = 0.0043$, Student's $t$-test). Black bars exemplify shRNA Ctrl and white bars shRNA $Usp9X$ samples.

C   Immunoblot analysis of HeLa cells transfected with a FLAG-tagged USP9X expression construct or empty vector (EV) and treated with taxol for the indicated times.

D   Immunoblot analysis of HeLa cells transfected with a FLAG-tagged XIAP expression construct or empty vector and treated as in (C).

E   Two-dimensional cell cycle analysis (BrdU/PI) of cells described in (C) and (D). Apoptotic indices represent ratios of sub-G1 to G1/S cells and are shown for analyses at the indicated time points ($n = 3$, $\pm$ SD). **$P = 0.0020$; ***$P = 0.0004$; Student's $t$-test.

F   Immunoblot analysis of HeLa cells that were transfected with siRNA oligonucleotides directed against a non-relevant mRNA (Ctrl) or against $Xiap$ mRNA and treated with taxol as specified.

Source data are available online for this figure.

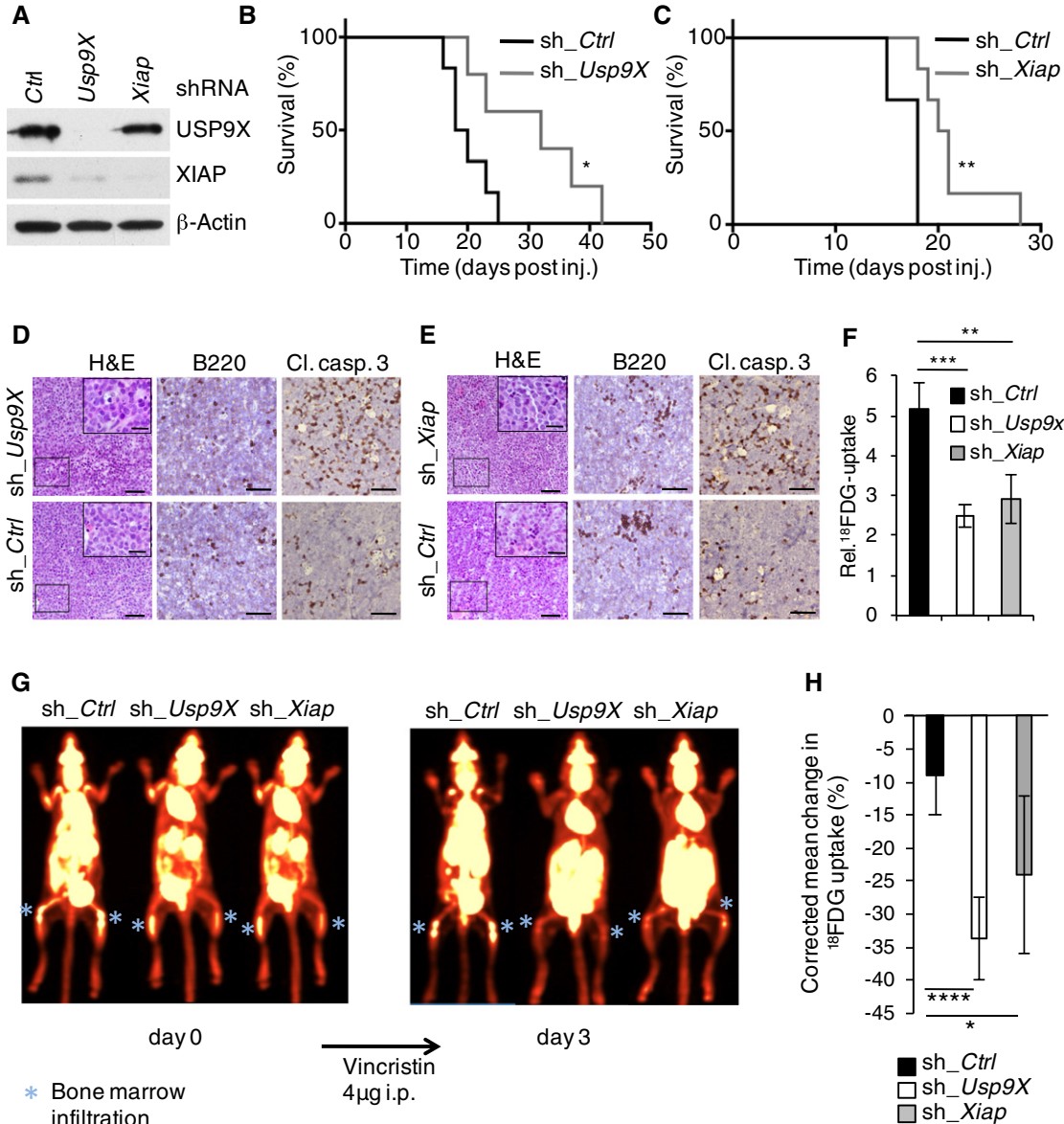

**Figure 3. USP9X and XIAP promote lymphoma growth and resistance to spindle poisons *in vivo*.**

A   Immunoblot analysis of sorted Eμ-Myc lymphoma cells that were retrovirally transduced with the indicated IRES-GFP shRNA constructs.

B   Kaplan–Meier survival curves of mice that were injected intravenously with syngeneic Eμ-Myc lymphoma cells transduced with the indicated shRNA constructs as described in (A). Prior to injection, cells were sorted for viability (propidium iodide negative) and GFP positivity (sh_Ctrl, black, *n* = 6; sh_*USP9X*, gray, *n* = 5). *\*P* = 0.0293; Mantel–Cox test.

C   Kaplan–Meier survival curves of mice that were injected with syngeneic Eμ-Myc lymphoma cells transduced with the indicated shRNA constructs as described in (A) and treated as in (B) (sh_Ctrl, black, *n* = 6; sh_*XIAP*, gray, *n* = 6). *\*\*P* = 0.0020; Mantel–Cox test.

D   Histopathology of representative lymph nodes derived from mice shown in (B) to visualize histomorphology by hematoxylin/eosin (H&E), B-cell origin (B220), and spontaneous apoptosis (cleaved caspase-3) *in situ*. Scale bars denote 50 μm and 20 μm for insets.

E   Histopathology of representative lymph nodes derived from mice shown in (C) which were stained as described in (D). Scale bars denote 50 μm and 20 μm for insets.

F   Mean tumor burden in animals injected with Eμ-Myc lymphoma cells modified as in (B) and (C), 15 days after injection, assessed by PET imaging (four animals per group) (± SD). *\*\*P* = 0.0025; *\*\*\*P* = 0.0003, Student's *t*-test.

G   Representative PET images of animals, injected with Eμ-Myc lymphoma cells modified as in (B) and (C), 15 days after injection (pre-treatment) and 3 days following intraperitoneal vincristine administration (post-treatment). Bone marrow infiltration, representative for systemic lymphoma manifestation, is indicated by a blue asterisk.

H   Mean change in standardized ¹⁸FDG uptake of ROI between pre- and post-treatment imaging corrected for initial tumor burden (sh_Ctrl, black, *n* = 6; sh_*USP9X*, white, *n* = 8; sh_*Xiap*, gray, *n* = 8). Within the sh_*Usp9X* and sh_*Xiap* groups, all animals lived up to control imaging, while two of eight animals of the sh_Ctrl group had to be sacrificed prematurely because they had reached the predefined criteria of maximum tumor burden. These animals had to be excluded from final analysis, suggesting an underestimation of the actual effect. Data were obtained from two independent experiments (± SD). *\*\*\*\*P* < 0.0001; *\*P* = 0.0167, Student's *t*-test

Source data are available online for this figure.

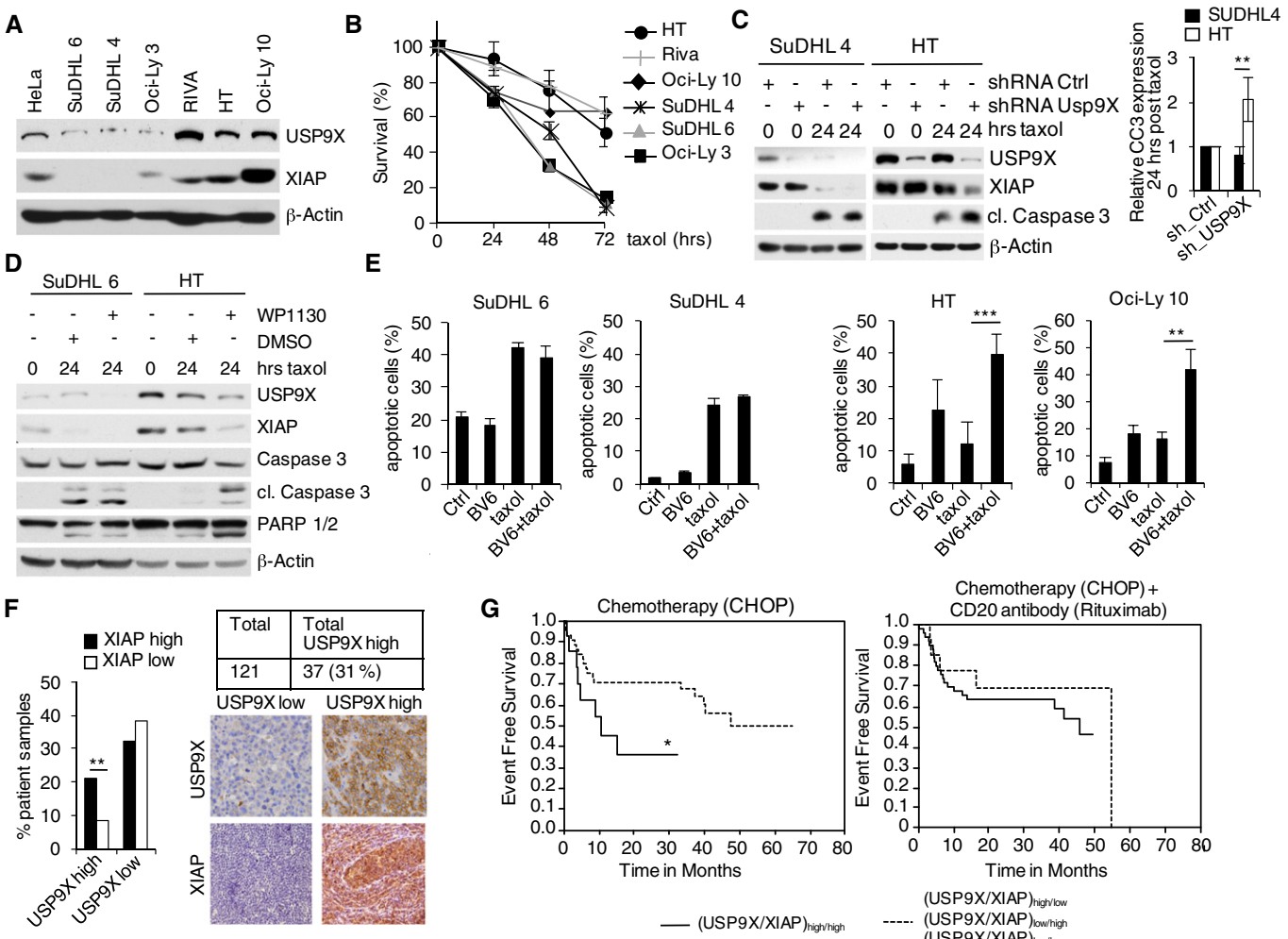

**Figure 4. USP9X is overexpressed in DLBCL to promote survival and resistance to anti-tubulin chemotherapeutics via mitotic stabilization of XIAP.**

A   Immunoblot analyses of different human DLBCL lines using antibodies to the indicated proteins.

B   FACS analysis (propidium iodide, PI) of USP9X high- and low-expressing DLBCL cells shown in (A) to determine cell survival. Cells were treated with taxol and collected at the specified time points. PI-negative cells are indicated relative to time point 0 ($n = 3$, $\pm$ SD).

C   Immunoblot analyses of the indicated DLBCL cell lines that were lentivirally transduced with IRES-GFP shRNA constructs against *USP9X* or a non-relevant mRNA, FACS sorted for GFP$^+$ PI$^-$ cells and exposed to taxol for the indicated periods of time. The quantification of cleaved caspase-3 (CC3) at the 24 h post-taxol treatment time point from four independent experiments is quantified in the graph on the right side and normalized to control shRNA ($\pm$ SD). **$P = 0.0099$, Student's *t*-test.

D   Immunoblot analyses of the indicated DLBCL cell lines that were exposed to taxol for the indicated periods of time. Three hours before collecting, WP1130 at a concentration of 5 μM or DMSO was added as specified.

E   FACS analysis (propidium iodide (PI) uptake) of DLBCL cell lines treated with taxol and/or the SMAC mimetic BV6 as indicated. Results displayed are from three independent experiments ($n = 3$, $\pm$ SD). ***$P = 0.00085$; **$P = 0.00529$; Student's *t*-test.

F   Collective assessment of USP9X and XIAP expression determined by immunohistochemistry in tissue microarrays (TMA) derived from a patient cohort representing 121 evaluable human CD20$^+$ aggressive B-cell lymphomas. High USP9X expression was identified in 37 cases (31%) and significantly correlated with high XIAP expression levels (**$P = 0.004$; Student's *t*-test).

G   Expression data derived from (E) were correlated with clinical follow-up. Within a cohort of 58 patients receiving vincristine-containing chemotherapy (CHOP regimen: cyclophosphamide, doxorubicin, vincristine, prednisone) without the addition of the anti-CD20 antibody rituximab (left panel), overexpression of USP9X and XIAP ($n = 14$, straight line) was associated with a significantly shortened event-free survival (EFS) as compared to all remaining patients ($n = 44$, dotted line) (*$P = 0.050$; log-rank test). Within a cohort of 63 patients receiving the CHOP regimen with the addition of the anti-CD20 antibody rituximab, overexpression of USP9X and XIAP ($n = 14$, straight line) was not associated with a shortened event-free survival (EFS) as compared to all remaining patients ($n = 49$, dotted line) ($P = 0.454$).

Source data are available online for this figure.

potentiation of taxol-induced cell death in USP9X/XIAP high/taxol-insensitive (HT, Oci-Ly 10, RIVA) cells, while this effect was not present in USP9X/XIAP low/taxol-sensitive (SuDHL 4, SuDHL 6, Oci-Ly 3) cells (Figs 4E and EV3C). SMAC mimetics have been shown to induce degradation of cIAP1/2 (Li *et al*, 2004;

Varfolomeev *et al*, 2007; Vince *et al*, 2007). To rule out the possibility that increased sensitivity of USP9X/XIAP high-expressing cells to BV6 results from increased cIAP1/2 degradation, we analyzed cIAP1/2 levels in BV6-treated SuDHL 6 and HT cells. Indeed, USP9X/XIAP high-expressing HT cells demonstrated only marginal

changes of cIAP1/2, in explicit contrast to USP9X/XIAP low-expressing SuDHL 6 cells, while XIAP was readily destabilized (Fig EV3D). These findings further underscore a role of XIAP as the relevant USP9X substrate for mitotic survival in B-cell lymphoma.

To further extend our studies on the role of the USP9X–XIAP axis in aggressive human B-cell lymphoma, we analyzed USP9X/XIAP expression by immune histochemistry (IHC) in tissue microarrays (TMAs) assembled from a cohort of 121 human CD20$^+$ aggressive B-cell lymphoma samples. These specimens were derived from the RICOVER-60 trial of the German High Grade Non-Hodgkin's Lymphoma study Group (DSHNHL) and shown to be representative of the full study population (Appendix Table S1). In the RICOVER-60 trial, patients were randomized to receive six or eight cycles of a chemotherapy regimen consisting of cyclophosphamide, doxorubicin, vincristine, and prednisone (CHOP) alone or in combination with the CD20-specific antibody rituximab (Pfreundschuh *et al*, 2008). Indeed, IHC analyses revealed high expression of USP9X in 37 patients (31%), which significantly correlated with high XIAP expression ($P = 0.004$) (Fig 4F). We further evaluated whether high USP9X/XIAP expression correlates with clinical outcome in the patient cohort indicated above. When treated with CHOP without rituximab ($n = 58$), patients with USP9X/XIAP overexpression had a significantly lower 2-year event-free survival [36% (95% CI 9–64%)] in a univariate analysis compared to all other patients [71% (95% CI 57–84%; *$P = 0.050$; Fig 4G)]. By contrast, no significant difference was observed between the two groups when treated with CHOP and the anti-CD20 antibody rituximab [2-year event-free survival of 67% (95%-CI 43–94%) vs. 63% (95%-CI 50–77%) for all other patients ($P = 0.454$)] (Fig 4G). While rituximab's mechanism of action is still not fully understood, a large body of evidence suggests complement-mediated cytotoxicity (CMC) and antibody-dependent cellular cytotoxicity (ADCC) as major means, rather than checkpoint-induced apoptosis (Weiner, 2010).

In this analysis, USP9X/XIAP double high cases were compared to the remaining cases (including cases with low USP9X and high XIAP) because these cases reflect the constellation where elevated XIAP is predicted to result from mitotic USP9X-mediated stabilization of XIAP. Other USP9X-independent causes of elevated XIAP levels likely exist, which would however not reflect the situation in which DLBCL cells depend on the mitotic activity of the USP9X–XIAP axis. Likewise, elevated USP9X expression without concomitant XIAP stabilization would not reflect mitosis-specific activation of this new mechanism. This observation thus further argues for a specific role of the USP9X–XIAP axis in mediating resistance against checkpoint activating agents and spindle poisons.

Diffuse large B-cell lymphoma can be categorized into three subtypes termed germinal-center B-cell-like (GCB) DLBCL, activated B-cell-like (ABC) DLBCL, and primary mediastinal B-cell lymphoma (PMBL), according to their gene expression profiles, which depend on different oncogenic pathways (Frick *et al*, 2011). The ABC subtype has been associated with unfavorable prognosis and increased resistance to standard treatment strategies (Frick *et al*, 2011). We therefore evaluated a large DLBCL gene profiling dataset with regard to USP9X expression in different subtypes (Lenz *et al*, 2008). This analysis however did not reveal a significant difference between USP9X overexpression in the ABC- or GCB-type DLBCL (Appendix Fig S3), in line with our respective analysis in DLBCL cell lines, which also did not detect a specific pattern of USP9X

expression (ABC-type: RIVA, Oci-Ly 10, Oci-Ly 3; GCB-type: HT, SuDHL 4, SuDHL 6). We therefore suggest that USP9X/XIAP aberrations are not linked to either DLBCL subtype.

Together, our analyses in patient samples identify overexpression of USP9X/XIAP in a significant number of patients with CD20$^+$ aggressive B-cell lymphoma and provide initial evidence that overexpression of USP9X/XIAP may serve as a prognostic marker in patients treated with DNA damaging agents and spindle poisons to define a subgroup of high risk and chemotherapy refractory patients, independent of the ABC/GCB classification. These insights may mark a framework from which to further investigate therapeutic approaches targeting USP9X and XIAP in patients with CD20$^+$ aggressive B-cell lymphoma that display high USP9X/XIAP expression levels or spindle poison refractory disease. Our data further suggest that the addition of SMAC mimetics, which are currently under evaluation in early clinical trials, to spindle poison-containing chemotherapy regimens may be such an approach that warrants further investigation.

## Materials and Methods

### Cell culture and drug treatments

HeLa, Cos7, NIH 3T3, and HEK 293T were cultured in Dulbecco's modified Eagle's medium (DMEM) supplemented with 10% fetal FBS (HEK 293T: 10% BS) and 1% penicillin/streptomycin (P/S), and Phoenix Eco cells were kept in the same medium without P/S. Mouse embryonic fibroblasts (MEFs) were maintained in DMEM containing 20% FBS, 1% nonessential amino acids, and 1% P/S. *Mcl-1*$^{-/-}$ MEF, *Xiap*$^{-/-}$ MEF, and *cIap2*$^{-/-}$ MEF were obtained by a material transfer agreement from the Walter and Eliza Hall Institute, Melbourne, Australia. The DLBCL lines HT, RIVA, SuDHL 4, and SuDHL 6 were cultured in RPMI-1640 with 20% FBS and 1% P/S Oci-Ly 3 and Oci-Ly 10 in IMDM with 20% FBS, 1% P/S, and 0.1% β-mercaptoethanol. Eμ-Myc cells were cultured in RPMI-1640 medium supplemented with FBS 10%, nonessential amino acids 1%, β-mercaptoethanol 0.1%, and P/S 1%. Where indicated, the following drugs were used: MG132 (10 μM), thymidine 2 mM, nocodazole 400 ng/ml, CHX (100 μg/ml), paclitaxel (500 nM), WP1130 (10 nM unless otherwise specified), doxorubicin (1 μM), RO-3306 (9 μM). For synchronization in mitosis, HeLa cells were treated with thymidine, if applicable 24 h after siRNA transfection. Twenty-four hours later, thymidine was washed off and either nocodazole was added for another 12 h or RO-3306 was added for 16 h and then washed off, and cells were collected one hour later. All cells tested mycoplasma negative by a PCR detection method.

### Biochemical methods

Extract preparation, immunoprecipitation, and immunoblotting have previously been described (Bassermann *et al*, 2008; Fernandez-Saiz *et al*, 2013; Baumann *et al*, 2014). Cell lysis was performed in buffer containing glycerol 5%, Tris–HCl pH 7.5 50 mM, NaCl 250 mM, EDTA 1 mM, NP40 0.1%, MgCl$_2$ 5 mM, and protease inhibitors (PIN, TPCK, TLCK, NaVa, DTT, glycerol-bisphosphate). Lysis buffer for immunoprecipitation experiments contained Tris–HCl

50 mM, NaCl 150 mM, EDTA 1 mM, Triton X-100 1%, and protease inhibitors. After immunoprecipitation, beads were washed once in lysis buffer and three times in low salt buffer (Tris–HCl pH 7.5 50 mM, NaCl 100 mM, and $MgCl_2$ 50 mM). For assessment of DUB activity, HeLa cells were synchronized in S phase and released as described above. At the given time points, cells were harvested and lysed in DUB activity lysis buffer (Tris–HCl pH 7.4 50 mM, $MgCl_2$ 5 mM, sucrose 250 mM, DTT 1 mM, ATP 2 mM) for 20 min. Subsequently, equal amounts of total protein were incubated with human recombinant HA-ubiquitin-vinyl sulfone (Boston Biochem, # U-212) at a concentration of 5 μM or solvent at 37°C for 30 min and vortexed every 10 min. Thereafter, SDS was added to 1% samples boiled at 95°C for 5 min and then incubated at room temperature for another 5 min. Thereafter, HA immunoprecipitation was performed. Protein *in vitro* translations and GST protein purifications were essentially performed as described previously (Bassermann *et al*, 2008; Fernandez-Saiz *et al*, 2013; Baumann *et al*, 2014).

## Antibodies

The following antibodies were used: beta-actin (1:10,000, mouse), BIRC8 (mouse, Abnova # H00112401-B01), cIAP1/2, cleaved caspase-3-5A1E (1:300, rabbit, Cell Signaling #9664), CUL1 (1:500, mouse, Invitrogen #32-2400), cyclin A (1:1,000, mouse, Santa Cruz #sc-751), cyclin B1 (1:1,000, mouse, Cell Signaling #4138), cyclin E (1:1,000, mouse, kind gift of M. Pagano), FLAG (1:1,000, rabbit, Sigma #F7425), FLAG-M2 (1:1,000, mouse, Sigma #F3165), HA-16B12 (1:2,000, mouse, Covance #MMS-101P), human MCL1 (1:500, rabbit, BD Pharmingen #554103), murine MCL1 (1:1,000, rabbit, A&D Serotec AHP1249), ML-IAP (mouse, Santa Cruz #sc-71592), NIAP (rabbit, Abcam #ab25968), PARP1/2 (1:1,000, rabbit, Santa Cruz #sc-7150), pHH3 (S10) (1:300, rabbit, Cell Signaling #9701), PLK1 (1:500, rabbit, Invitrogen #33-1700), survivin (1:3,000, rabbit, R&D Systems #AF886), USP7 (rabbit, Bethyl Lab. #A300-033A), USP9X (1:4,000, rabbit, Bethyl Laboratories #A301-351A), USP10 (rabbit, Bethyl Lab. #A300-900A), USP24 (Proteintech Europe, #13126-1-AP), USP28 (rabbit, Bethyl Lab. #A300-898A), V5 (1:1,000, rabbit, Sigma #V8137), XIAP (1:1,000, mouse, BD Biosciences #610716), XIAP (1:1,000, rabbit, Cell Signaling #2042), and XIAP (1:1,000, R&D Systems #AF8221).

## Immunofluorescence

For immunofluorescence microscopy, cells were grown on cover slips prior to fixation in ice-cold methanol at −20°C for 10 min. Primary antibodies were used as outlined above. Secondary antibodies (Alexa Fluor 488 rabbit and Alexa Flour 594 goat, Life Technologies) were used at a dilution of 1:1,000. Antibodies were diluted in IF Buffer (0.5% Tween-20 in PBS) and fixed cells mounted with ProLong(R) Gold antifade reagent with DAPI (Life Technologies). Images were taken using the laser-scanning confocal microscope FluoView FV10i (Olympus).

## Transient transfections and lentiviral DNA transfer

Transient transfection was performed using $CaCl_2$ as before (Bassermann *et al*, 2008) and Lipofectamine 2000© according to the manufacturer's description. For lentiviral shRNA transfer into mouse embryonic fibroblasts, Phoenix Eco cell-derived media were applied in four spin infections with Polybrene 8 μg/ml.

## Plasmids, shRNA, and siRNAs

*USP9X* and *XIAP* siRNAs were obtained from Thermo Scientific Dharmacon® (Cat No. J-006099-09; Cat No. L-004098-00), and siRNA transfection was performed using HiPerFect© Transfection Reagent (Qiagen). shRNAs against *Xiap* in plko-based vectors were a kind gift from L. Nilsson. shRNAs directed against murine *Usp9X* targeted the following 21mer sequence: 5′-GAT GAG GAA CCT GCA TTT CCA-3′.

## Cell cycle analyses

For BrdU incorporation and DNA content analysis, cells were pulsed with 10 μM BrdU (BD Biosciences) for 120 min and stained according to a standard protocol thereafter. Two-dimensional flow cytometry was performed to detect fluorescein and PI. To quantify cell cycle distribution, the FlowJo software (Tree Star Inc, Stanford) was applied.

## Studies in mice

Eμ-Myc lymphoma cells were infected with lentiviral IRES-GFP shRNA constructs directed against a non-relevant mRNA or against *USP9X* mRNA or *XIAP* mRNA together with IRES-linked GFP, sorted 48 h thereafter, and injected into syngeneic C57BL (Harlan) recipient mice ($5 \times 10^4$ cells per mouse). Equally housed, fed, and bred female mice of same size and age (10 weeks) were thereby randomly distributed into experimental groups. Mice were examined weekly by lymph node palpation. Disease state was defined by the presence of palpable lymph nodes. Time point of sacrifice was determined by an independent member of the mouse facility with no insight into experimental design. To assess cell immunophenotype and GFP content, single-cell suspensions were obtained by passing lymph nodes and bone marrow through 70-μm cell strainers. In preparation for FACS and immunoblotting, erythrocytes were lysed using buffer containing $NH_4Cl$ 0.15 M, HEPES 0.02 M, and EDTA 0.1 M. Samples for immunohistochemistry (IHC) were fixed in 5% PFA. Survival data in each approach were analyzed using the Kaplan–Meier method, applying the log-rank (Mantel–Cox) test for statistical significance. Animals were censored from analyses when sacrificed for non-tumor reasons. For the survival and tumor growth studies, a group size of at least five animals per condition was chosen, which allowed the detection of twofold differences in survival with a power of 0.89, assuming a two-sided test with a significance threshold α of 0.05 and a standard deviation of less than 50% of the mean.

For treatment experiments, the mice of each group, regardless of clinical tumor signs, underwent PET analysis at day 15 after injection of Eμ-Myc lymphoma cells. Hence, they were injected with vincristine 4 μg intraperitoneally and re-imaged 3 days thereafter (day 18).

Experiments were performed in accordance with the local ethical guidelines and approved by the responsible regional authorities (Regierung von Oberbayern).

## PET data analysis

Analysis was performed using the Inveon Research workplace to semiquantitatively assess the accumulation of tracer in the tumor. Three-dimensional regions of interest (ROIs) were drawn manually around lesions, and a threshold algorithm (50% maximum intensity – minimum intensity) was applied to each ROI to obtain an intensity value from the area with the highest tumor activity. Mean values were taken from the lesions, and the same lesions in each mouse were analyzed pre- and post-treatment.

Background activity was determined by establishing two identically sized 3D ROIs in the spinal muscles below the kidneys.

For evaluation of tumor response to vincristine, mean change in $^{18}$FDG uptake was assessed and corrected for tumor burden at the time of treatment initiation.

## Immunohistochemistry and tumor samples

For immunohistochemistry, tissue sections were deparaffinized. Antigen retrieval was carried out by pressure cooking in citrate buffer (pH 6 or pH 9) for 10 min. USP9X or XIAP was detected after an overnight incubation at 4°C by the DAKO REAL Detection Kit (DAKO) according to the manufacturer's protocol. Immunohistochemical data were interpreted by a certified pathologist blinded to clinicopathological data. Slides were evaluated using the Axioplan 2 microscope (Zeiss, Goettingen, Germany). A semiquantitative score for was used to evaluate protein expression as follows: score 0: no expression, score 1: expression in < 10% of tumor cells, score 2: expression in > 10% of tumor cells, score 3: expression in > 50% of tumor cells. Tumor samples showing scores 0–1 were called USP9X/XIAP low, and samples exhibiting scores 2–3 were summarized as USP9X/XIAP high. All human specimens were processed with informed consent in compliance with the institutional review board at the Faculty of Medicine of the Technical University of Munich and conformed to the principles set out in the WMA Declaration of Helsinki and the Department of Health and Human Services Belmont Report.

## Normalization and quantification of protein levels

Protein concentrations of whole-cell extracts (WCE) were determined using a Bio-Rad DC protein assay (Lowry assay). For each experiment, equal amounts of WCE were separated by SDS–PAGE and analyzed by immunoblotting. Equal protein levels in each lane were confirmed by Ponceau S staining of the membrane and by immunoblotting proteins whose levels are typically not regulated in response to apoptosis or cell cycle progression (*e.g.*, CUL1, β-actin).

## Statistical analyses

An objective was to analyze the association of USP9X–XIAP overexpression with event-free survival (EFS) in elderly patients with CD20$^+$ aggressive B-cell lymphoma, according to primary histology. EFS was defined as time from randomization to disease progression, start of salvage treatment, additional (unplanned) treatments, relapse, or death from any cause.

The association of USP9X–XIAP overexpression and outcome was tested by the log-rank test and graphically displayed via

**The paper explained**

**Problem**

Diffuse large B-cell lymphoma (DLBCL) is the most common type of malignant lymphoma in adults, making up approximately 30% to 35% of all cases. While the majority of DLBCL patients respond well to initial chemotherapy, about half of the responders eventually relapse and about 10% are refractory to initial treatment. Factors predicting response to chemotherapy are not well characterized.

**Results**

We here identify USP9X as the deubiquitylase which mediates mitosis-specific stabilization of the anti-apoptotic protein XIAP. We show that upregulation of USP9X and consequently of XIAP promote mitotic survival and increased resistance to mitotic spindle poisons in diffuse large B-cell lymphoma. Inactivation of both proteins sensitizes DLBCL cells to spindle poison-induced apoptosis. Furthermore, patients with high USP9X/XIAP expression demonstrate significantly reduced event-free survival after chemotherapeutic treatment.

**Impact**

Overexpression of USP9X and XIAP is identified as a predictive biomarker for chemotherapy resistance in aggressive B-cell lymphoma. Targeting of USP9X or XIAP in USP9X/XIAP-overexpressing and/or chemotherapy refractory tumors may be a promising approach to improve survival of DLBCL patients.

Kaplan–Meier curves. Two-year rates with 95% confidence intervals and *P*-values are presented. For comparison of patient characteristics for the groups with and without TMA material, Pearson's chi-square tests were performed. For comparison of the median age, Mann–Whitney *U*-test was used. A statistically significant difference was assumed for $P \leq 0.05$. Statistical analyses were done with SPSS statistics version 20.

Other statistical analysis of results was performed by log-rank (Mantel–Cox) test, Student's *t*-test, or one-way ANOVA, as specified in the individual figure legends according to the assumptions of the test using GraphPad Prism software. The bars shown represent the mean ± standard deviation (SD). Murine outcome data were visualized by Kaplan–Meier curves, and survival analyses were performed using a log-rank test. The *P*-values are presented in figure legends where a statistically significant difference was found. $*P < 0.05$; $**P < 0.01$; $***P < 0.001$.

**Expanded View** for this article is available online.

## Acknowledgements

We thank V. Dixit, J. Hastie, L. Nilsson, and S. Wood, for reagents and cell lines, and M. Yabal for suggestions. This work was supported by fellowships from the TU München (KKF B07-11) and the Fritz-Thyssen Foundation to K.E. and grants from the German Research Foundation (KE 222/7-1 and SFB 824 to U.K. and BA 2851/4-1 and SFB 1243 to F.B.), the German Cancer Aid (#111430), and the Wilhelm Sander Stiftung (#2012.096.1) to F.B.

## Author contributions

KE and FB conceived and designed the research; KE performed most of the experiments with crucial help from AR, JS, JK, AB, B-ST, FL, and VF-S; MP, LT, and WK provided the lymphoma samples which were analyzed by MR, BA, and AR; BAA and SF performed experiments using SMAC mimetics; KE, LJ, MR, BA,

CJG, MU, VF-S, MP, LT, WK, SF, UK, PJ, CP, and FB analyzed results; FB coordinated this work and wrote the manuscript together with KE All authors discussed the results and commented on the manuscript.

## Conflict of interest

The authors declare that they have no conflict of interest.

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
