## [Review Process File · EMBO Molecular Medicine]

USP9X stabilizes XIAP to regulate mitotic cell death and chemoresistance in aggressive B-cell lymphoma

Katharina Engel, Martina Rudelius, Jolanta Slawska, Laura Jacobs, Behnaz Ahangarian Abhari, Bettina Altmann, Julia Kurutz, Abirami Rathakrishnan, Vanesa Fernandez-Sáiz, Andrä Brunner, Bianca-Sabrina Targosz, Felicia Loewecke, Christian Johannes Gloeckner, Marius Ueffing, Simone Fulda, Michael Pfreundschuh, Lorenz Trümper, Wolfram Klapper, Ulrich Keller, Philipp J. Jost, Andreas Rosenwald, Christian Peschel, and Florian Bassermann

Corresponding author: Florian Bassermann, Technical University of Munich

Review timeline:	Submission date:	10 November 2015
	Editorial Decision:	10 December 2015
	Revision received:	13 April 2016
	Editorial Decision:	26 April 2016
	Revision received:	04 May 2016
	Accepted:	17 May 2016

Transaction Report:

Editor: Roberto Buccione

1st Editorial Decision 10 December 2015

Thank you for the submission of your manuscript to EMBO Molecular Medicine. We are sorry that it has taken longer than usual to get back to you on your manuscript. In this case we experienced some difficulties in securing three appropriate expert reviewers and then obtaining their evaluations in a timely manner.

We have now received comments from the two out of the three Reviewers whom we asked to evaluate your manuscript. To avoid further delays, I am proceeding based on the two evaluations of Reviewers 1 and 2.

As you will see, both reviewers find the manuscript interesting, but raise a significant number of concerns centred on conclusiveness and quality of the data in general, some of which also impinge on clinical relevance. These issues, in part overlapping, are complementary and well taken.

While publication of the paper cannot be considered at this stage, we would be pleased to consider a suitably revised submission, provided, however, that the Reviewers' concerns are fully addressed with further experimentation where required.

Please note that it is EMBO Molecular Medicine policy to allow a single round of revision only and that, therefore, acceptance or rejection of the manuscript will depend on the completeness of your responses included in the next, final version of the manuscript.

As you know, EMBO Molecular Medicine has a "scooping protection" policy, whereby similar findings that are published by others during review or revision are not a criterion for rejection. However, I do ask you to get in touch with us after three months if you have not completed your revision, to update us on the status. Please also contact us as soon as possible if similar work is published elsewhere.

EMBO Molecular Medicine now requires a complete author checklist (<http://embomolmed.embopress.org/authorguide#editorial3>) to be submitted with all revised manuscripts. Provision of the author checklist is mandatory at revision stage; The checklist is designed to enhance and standardize reporting of key information in research papers and to support reanalysis and repetition of experiments by the community. The list covers key information for figure panels and captions and focuses on statistics, the reporting of reagents, animal models and human subject-derived data, as well as guidance to optimise data accessibility.

I look forward to seeing a revised form of your manuscript as soon as possible.

***** Reviewer's comments *****

Referee #1 (Comments on Novelty/Model System):

Some parts are technically not solid (for example protein expression levels are not comparable) and the *in vivo* experiments seem to have been performed only once. Also, some conclusions are derived from a very small number of cell lines that are not compared systematically. Therefore I find the technical quality rather weak.

Referee #1 (Remarks):

The manuscript by Engel et al. addresses the role of the deubiquitinase USP9X in the regulation of XIAP stability and chemoresistance of lymphoma cells. The authors show that manipulation of USP9X levels can lead to a correlating change in XIAP ubiquitination and protein levels, and that this is possibly mediated by USP9X-mediated deubiquitination of XIAP. Moreover, the authors propose that cancer cells with high USP9X levels have a reduced sensitivity to mitosis-inhibiting anti-cancer reagents such as the microtubule inhibitor taxol. In a mouse model of xenografted, Myc-induced lymphoma, knock-down of USP9X or XIAP indeed lowered the progression of these lymphomas. In principle, these findings are highly interesting and relevant to our understanding of the proto-oncogenic function of XIAP. However, I have major concerns about the technical quality of some of the data and I also disagree with some of the authors' interpretations.

Major concerns:

1) Fig. 1D: the authors' conclusions are not fully supported by the data; the amounts of mutant XIAP constructs in the IPs seem to be considerably lower than for the wt, therefore reduced USP9X binding could be simply the result of lower XIAP levels in the IP.

2) Fig. 1F: same problem as in Fig. 4D. Expression levels of mutant USP9X are considerably lower, therefore no valid conclusion can be drawn about the relevance of USP9X enzymatic activity for XIAP stabilization.

3) Fig. 1G: the stabilizing effect of transfected USP9X on XIAP is not very convincing, it would be much more pertinent to assess the effect of WP1130 on endogenous USP9X and XIAP in HeLa cells (that endogenous USP9X is inhibited is indeed suggested by the fact that XIAP levels drop down to below levels of untransfected cells at high WP1130 concentrations)

4) Data in Figure 3 seem to be derived from a single experiment, it is therefore unclear how reproducible these findings are. A second experiment should be performed and mean values and standard deviations should be included in Fig. 3G.

5) Based on data shown in Figure 4, the authors conclude that DLBCL with high versus low USP9X/XIAP expression levels show a correlating sensitivity to taxol treatment, however the data are based on a very limited number of cell lines, which does not allow to draw such conclusions, especially since the absolute differences in USP9X protein levels are not strong. To strengthen their conclusions, the authors must use a higher number of cell lines to start with, and perform experiments such as those performed in Fig. 4C and 4D in several cell lines (not just one of each, and the same in all figure panels). The resulting sensibility of the cell lines to taxol treatment must be correlated with data shown in Fig. 4C and 4D. The authors should also assess whether USP9X and XIAP levels systematically differ between DLBCL cell lines of the GCB and ABC subtype. This is particularly important because DLBCL of the ABC subtype are known to respond much less well to CHOP therapy.

6) In Fig. 4G, it is not clear why the authors compare double high cases with a second group including XIAP high patients. How would the results look if comparing USP9X high versus low cases? And XIAP high versus XIAP low cases? Please explain and comment in text.

Minor points:

1) It is important to mention the existence of ABC and GCB subtypes of DLBCL and to discuss whether these subtypes may differ in their USP9X and XIAP expression levels, since ABC and GCB DLBCL are known to respond differently to CHOP chemotherapy.

2) Fig. 3B,C: please check whether USP9X or XIAP silencing affects the in vitro viability of the cell lines.

3) SUDHL8 cell lines are included in Fig. 4A but results are not commented and cell line was not further tested

4) Fig. 1, co-immunoprecipitation of USP9X and XIAP was demonstrated only for overexpressed proteins or in a semi-endogenous manner, this should be explicitly mentioned in the results section

Referee #2 (Comments on Novelty/Model System):

This study by Engel et al. sets out to investigate a hypothesised role for deubiquitinases (DUBs) in regulating APC/C function and mitosis controlled by the spindle-assembly check point (SAC). The rationale is that although APC/C is a well-described ubiquitin E3 ligase in SAC signalling, the regulation of these processes by DUBs remains poorly understood. The authors identify USP9x as a DUB enriched in mitotic cells and show that USP9x regulates XIAP levels during mitosis. This, in turn, impinges on cell viability and sensitivity to spindle poison drugs. Correlating with their cell culture experiments, the authors find that B-cell lymphomas with high USP9x and XIAP are more resistant to treatment and patients with high level of these proteins have poor prognosis relative to other patients. They conclude that USP9x/XIAP levels could be a valuable prognostic marker for

this cancer.

This is an interesting study that provides novel insight into the biology of B-cell lymphoma. The experimental layout is logical and the conclusions are by-in-large supported by the data.

Referee #2 (Remarks):

This study by Engel et al. sets out to investigate a hypothesised role for deubiquitinases (DUBs) in regulating APC/C function and mitosis controlled by the spindle-assembly check point (SAC). The rationale is that although APC/C is a well-described ubiquitin E3 ligase in SAC signalling, the regulation of these processes by DUBs remains poorly understood. The authors identify USP9x as a DUB enriched in mitotic cells and show that USP9x regulates XIAP levels during mitosis. This, in turn, impinges on cell viability and sensitivity to spindle poison drugs. Correlating with their cell culture experiments, the authors find that B-cell lymphomas with high USP9x and XIAP are more resistant to treatment and patients with high level of these proteins have poor prognosis relative to other patients. They conclude that USP9x/XIAP levels could be a valuable prognostic marker for this cancer.

This is an interesting study that provides novel insight into the biology of B-cell lymphoma. The experimental layout is logical and the conclusions are by-in-large supported by the data. However, I do have some concerns/suggestions that should be addressed:

- 1) DUBs are not necessarily always active and it would be relevant to address USP9x activity through the cell cycle and not only its protein levels. It would thus strengthen the study to demonstrate that also USP9x activity is elevated in mitosis. This could be investigated using DUB activity probes (e.g. HA-Ub-vinyl sulfone, Boston Biochem). Similar reagents are available from UbiQ. See also McGouran et al. Chem Biol. 2014.
- 2) I am not sure it can be claimed that the XIAP - USP9x interaction is direct as stated on page 4 "we tested direct interaction of USP9X and XIAP". The interaction might involve one or more other proteins. To address this, the authors should test the interaction in vitro with purified protein or modify the statement. In any case, it would be interesting to determine the mode of interaction in more detail. Relating to this, the G188 residue in BIR2 is not surface exposed but is positioned within the core of the domain and mutation to a charged residue will likely destabilize the entire domain. The interpretation of the USP9x binding results should consider this. It would be worth testing if Smac or Smac mimetic compounds that interfere with other BIR2-binding partners (e.g. RIP2; Krieg et al. PNAS 2009, Damgaard et al. 2013) would interfere with USP9x binding. Has this been tested?
- 3) Smac mimetics such as BV6 degrade cIAP1/2 rapidly. Could the differential sensitivity of SuDHL and HT cells to Taxol + BV6 be related to different cIAP1/2 levels in these cells?
- 4) The survival time of mice injected with lymphoma cells suggest to me that USP9x functions also independently of XIAP to promote survival/growth since the effect of USP9x silencing was greater than when XIAP was silenced (Fig 3B and 3C). It would be of interest to study the effects when silencing both XIAP and USP9x. Would the effect be additive or be similar to USP9x silencing?

1st Revision - authors' response

13 April 2016

I am enclosing a revised version of our manuscript (Engel et al., EMM-2015-06047) titled "USP9X stabilizes XIAP to regulate mitotic cell death and chemoresistance in aggressive B-cell lymphoma" in which we have addressed all specific and general issues raised by the referees.

Based on the various helpful comments by the reviewers, we have now conducted a significant number of additional experiments and validations. These analyses provide further insights into the

mechanism by which USP9X interacts with XIAP to mediate its mitosis-specific deubiquitylation and stabilization. In addition, we further delineate how aberrant activation of this pathway promotes growth, survival and treatment resistance of aggressive B-cell lymphoma using syngeneic murine B-cell lymphoma models and different cell culture models of diffuse large B-cell lymphoma (DLBCL). Finally, we further validate our claim that high USP9X/XIAP expression levels may serve as a prognostic biomarker to define a subgroup of high risk and chemotherapy refractory DLBCL patients, independent of the ABC/GCB subtype classification.

Please find below our detailed point-by-point response to the reviewers' comments.

Reviewer #1:

This reviewer believes that our findings are important and relevant. He/she states "In principle, these findings are highly interesting and relevant to our understanding of the proto-oncogenic function of XIAP." However, this reviewer has some concerns with regard to the technical quality of some of the data as well as some interpretations and has therefore asked us to address the following specific issues (italicized):

Major concerns:

"1) Fig. 1D: the authors conclusions are not fully supported by the data; the amounts of mutant XIAP constructs in the IPs seem to be considerably lower than for the wt, therefore reduced USP9X binding could be simply the result of lower XIAP levels in the IP."

We agree with the reviewer that the previous experiment was of limited significance, given the different levels of precipitated mutant and wt XIAP. We therefore repeated this experiment under conditions in which equal amounts of either mutant or wt forms of XIAP were precipitated and now clearly demonstrate that both XIAP mutants (XIAPG188E and XIAPG188R) do not interact with USP9X, in explicit contrast to wt XIAP (new Fig. 1D).

"2) Fig. 1F: same problem as in Fig. 4D. Expression levels of mutant USP9X are considerably lower, therefore no valid conclusion can be drawn about the relevance of USP9X enzymatic activity for XIAP stabilization."

We repeated this experiment under conditions of equal expression of wt USP9X and the catalytically inactive USP9X mutant (USP9XC1556S). The new Fig. 1F now clearly demonstrates that the stabilizing effect of USP9X on XIAP is dependent on its enzymatic activity.

"3) Fig. 1G: the stabilizing effect of transfected USP9X on XIAP is not very convincing, it would be much more pertinent to assess the effect of WP1130 on endogenous USP9X and XIAP in HeLa cells (that endogenous USP9X is inhibited is indeed suggested by the fact that XIAP levels drop down to below levels of untransfected cells at high WP1130 concentrations)".

As suggested by the reviewer, we repeated this experiment without transfected USP9X. We now demonstrate that WP1130 dramatically decreases endogenous XIAP protein expression (new Fig. 1G).

"4) Data in Figure 3 seem to be derived from a single experiment, it is therefore unclear how reproducible these findings are. A second experiment should be performed and mean values and standard deviations should be included in Fig. 3G."

As requested, we repeated the treatment experiment shown in the previous Fig. 3F, G. We now reproduce our previous results and show that silencing of USP9X and XIAP sensitizes aggressive lymphoma to vincristine treatment using the syngeneic E μ -MYC model and FDG-PET imaging. Respective mean values and standard deviations from six (sh_Ctrl) or eight (sh_Usp9X and sh_Xiap) animals per group are shown in the new Fig. 3H. In addition, our PET analyses again reveal significantly reduced metabolic activity of tumors derived from USP9X and XIAP E μ -MYC cells, supporting the data shown in Fig. 3B-E (new Fig. 3F). Of notice, for the experiments shown in

Fig. 3B-E, we chose a group size of at least 5 animals per condition, which allowed the detection of two fold differences in survival with a power of 0.89, assuming a two-sided test with a significance threshold of 0.05 and a standard deviation of less than 50% of the mean.

“5) Based on data shown in Figure 4, the authors conclude that DLBCL with high versus low USP9X/XIAP expression levels show a correlating sensitivity to taxol treatment, however the data are based on a very limited number of cell lines, which does not allow to draw such conclusions, especially since the absolute differences in USP9X protein levels are not strong. To strengthen their conclusions, the authors must use a higher number of cell lines to start with, and perform experiments such as those performed in Fig. 4C and 4D in several cell lines (not just one of each, and the same in all figure panels). The resulting sensibility of the cell lines to taxol treatment must be correlated with data shown in Fig. 4C and 4D.”

We agree with the reviewer that the indicated experiments should be performed in more cell lines in order to strengthen our conclusions. We therefore took great effort to address this issue. We now present the data of previous Fig. 4C (silencing of USP9X in USP9X/XIAP high and low cells) in two low (SuDHL4, SuDHL6) and two high (HT, RIVA) DLBCL lines, the data of previous Fig. 4D (treatment with WP1130 in USP9X/XIAP high and low cells) in two low (SuDHL6, Oci-Ly3) and two high (HT, Oci-Ly10) DLBCL lines, and the data of previous Fig. 4E (treatment with the SMAC mimetic BV6 +/- taxol in USP9X/XIAP high and low cells) in three low (SuDHL6, SuDHL4, Oci-Ly3) and three high (HT, Oci-Ly10, RIVA) DLBCL lines (New Fig. 4C, D, E; Fig. EV 3A, B, C). USP9X/XIAP expression and the taxol sensitivity of these cells is shown in new Fig. 4A, B). Importantly, these experiments revealed correlating results in each individual setting, thus underscoring our claim that USP9X stabilizes XIAP in mitosis to mediate resistance to spindle poisons in DLBCL.

“The authors should also assess whether USP9X and XIAP levels systematically differ between DLBCL cell lines of the GCB and ABC subtype. This is particularly important because DLBCL of the ABC subtype are known to respond much less well to CHOP therapy.”

This is indeed an important point. Our respective analysis however did not detect a specific pattern of USP9X expression among these DLBCL subtypes (ABC-type: RIVA, Oci-Ly 10, Oci-Ly 3; GCB-type: HT, SuDHL 4, SuDHL 6). We additionally evaluated a large DLBCL gene profiling dataset with regard to USP9X expression in different subtypes (Lenz et al., NEJM 2008). This analysis also did not reveal a significant difference between USP9X overexpression in either subtype (Appendix Fig. S3). We therefore suggest that USP9X/XIAP aberrations are not linked to a DLBCL subtype.

“6) In Fig. 4G, it is not clear why the authors compare double high cases with a second group including XIAP high patients. How would the results look if comparing USP9X high versus low cases? And XIAP high versus XIAP low cases? Please explain and comment in text.”

We compared the double high cases to the remaining cases (including cases with low USP9X and high XIAP) because these cases reflect the constellation where elevated XIAP is predicted to result from mitotic USP9X-mediated stabilization of XIAP. Other USP9X-independent causes of elevated XIAP levels likely exist, which would however not reflect the situation in which DLBCL cells depend on the mitotic activity of the USP9X-XIAP axis. Likewise, elevated USP9X expression without concomitant XIAP stabilization would not reflect mitosis-specific activation of this new mechanism. Accordingly, when comparing USP9X and XIAP high versus low cases independently of concomitant levels of XIAP and USP9X respectively, no significant differences between these groups were observed (see below). This data was obtained from the same patient cohort as described in our manuscript. This finding further supports our claim that mitotic stabilization of XIAP marks an important means by which USP9X promotes lymphoma growth and treatment resistance. This context is now discussed in the manuscript (page 9).

Minor points:

“1) It is important to mention the existence of ABC and GCB subtypes of DLBCL and to discuss whether these subtypes may differ in their USP9X and XIAP expression levels, since ABC and GCB DLBCL are known to respond differently to CHOP chemotherapy.”

As specified under point 5 above, we evaluated a large DLBCL gene profiling dataset with regard to USP9X expression in different subtypes (Lenz et al., *N Engl J Med* (2008) 359: 2313-2323) (Appendix Fig. S3), and examined our DLBCL cell lines with regard to their subtype status. These studies however did not reveal a significant difference between USP9X overexpression in either subtype. We therefore conclude that USP9X/XIAP aberrations are not linked to a DLBCL subtype.

We have included a paragraph in the manuscript in which we mention the existence of ABC and GCB subtypes of DLBCL and discuss our respective findings (page 9 of the manuscript).

“2) Fig. 3B,C: please check whether USP9X or XIAP silencing affects the in vitro viability of the cell lines.”

We performed the requested analysis and show that USP9X or XIAP silencing indeed affects in vitro viability of the Eμ-Myc cells exposed to Taxol used in Fig. 3B, C (new Appendix Fig. S2A). However, it is important to indicate that FACS sorting for PI negativity was performed immediately before injection of lymphoma cells and therefore only viable cells were applied in all groups.

“3) SUDHL8 cell lines are included in Fig. 4A but results are not commented and cell line was not further tested”

We did not perform any functional analyses with SUDHL8 cells. We have therefore removed these cells from Fig. 4A.

“4) Fig. 1, co-immunoprecipitation of USP9X and XIAP was demonstrated only for overexpressed proteins or in a semi-endogenous manner, this should be explicitly mentioned in the results section”

We now also specify the conditions of USP9X and XIAP co-immunoprecipitations in the results section (page 4).

Reviewer #2:

This reviewer appears very enthusiastic about our study and states: “This is an interesting study that provides novel insight into the biology of B-cell lymphoma. The experimental layout is logical and the conclusions are by-in-large supported by the data.”

She/he has asked us to address the following points:

“1) DUBs are not necessarily always active and it would be relevant to address USP9x activity through the cell cycle and not only its protein levels. It would thus strengthen the study to demonstrate that also USP9x activity is elevated in mitosis. This could be investigated using DUB activity probes (e.g. HA-Ub-vinyl sulfone, Boston Biochem). Similar reagents are available from UbiQ. See also McGouran et al. Chem Biol. 2014.”

We would like to thank the reviewer for this experimental suggestion. We performed the experiment as requested and indeed find elevated USP9X activity in mitosis, thus further underscoring our claim that USP9X deubiquitylates and stabilizes XIAP in mitosis (new Fig EV 2B).

“2) I am not sure it can be claimed that the XIAP - USP9x interaction is direct as stated on page 4 "we tested direct interaction of USP9X and XIAP". The interaction might involve one or more other proteins. To address this, the authors should test the interaction in vitro with purified protein or modify the statement.”

We performed the requested experiment, using GST-purified XIAP and in-vitro translated and 35S-labelled USP9X. For USP9X, different fragments which cover the full USP9X protein were used, given the large size of the protein (>250 kDa) which does allow in-vitro translation of the full protein. We now demonstrate specific binding of USP9X to XIAP in the setting of purified proteins, suggesting that the interaction is indeed direct (New Fig. EV 1A).

“In any case, it would be interesting to determine the mode of interaction in more detail. Relating to this, the G188 residue in BIR2 is not surface exposed but is positioned within the core of the domain and mutation to a charged residue will likely destabilize the entire domain. The interpretation of the USP9x binding results should consider this. It would be worth testing if Smac or Smac mimetic compounds that interfere with other BIR2-binding partners (e.g. RIP2; Krieg et al. PNAS 2009, Damgaard et al. 2013) would interfere with USP9x binding. Has this been tested? “

To address this issue, we investigated whether SMAC mimetics interfere with the binding of USP9X to XIAP, as requested. Indeed, we found that the SMAC mimetic BV6 disrupts binding of both proteins, suggesting that binding occurs via the BIR2 of XIAP and further underscoring our finding that binding is mediated via the G188 residue (new Fig. EV 1D).

“3) Smac mimetics such as BV6 degrade cIAP1/2 rapidly. Could the differential sensitivity of SuDHL and HT cells to Taxol + BV6 be related to different cIAP1/2 levels in these cells?”

As requested, we analyzed cIAP1/2 levels in BV6-treated SuDHL6 and HT cells. USP9X/XIAP high expressing HT cells demonstrated only marginal changes of cIAP1/2, in explicit contrast to USP9X/XIAP low expressing SuDHL6 cells (Fig. EV 3D). This finding further underscores our claim that the differential sensitivity of SuDHL and HT cells to Taxol + BV6 relates to differences in XIAP expression.

“4) The survival time of mice injected with lymphoma cells suggest to me that USP9x functions also independently of XIAP to promote survival/growth since the effect of USP9x silencing was greater than when XIAP was silenced (Fig 3B and 3C). It would be of interest to study the effects when silencing both XIAP and USP9x. Would the effect be additive or be similar to USP9x silencing?”

In response to this point, we analyzed survival of mice receiving Eμ-MYC cells with simultaneous knockdown of both Usp9X and Xiap. Notably, double knockdown of Usp9X and Xiap did not increase survival as compared to the single Xiap knockdown, indicating that USP9X functions via XIAP to promote lymphoma survival and growth in this model (Appendix Fig. S2D, E).

I sincerely appreciate the constructive suggestions made by the reviewers that improved the clarity and message of the paper.

I am looking forward to hearing from you.

2nd Editorial Decision

26 April 2016

Thank you for the submission of your manuscript to EMBO Molecular Medicine. We have now heard back from the two reviewers whom we asked to evaluate your manuscript.

As you will see, while Reviewer #2 is now satisfied, Reviewer 1 raises important concerns.

These issues prevent us from moving forward with your manuscript and we must therefore ask you to take action on the following:

1) Reviewer 1 notes that in the revised Fig. 1D, which should have featured new data, the blots for the IP (blot anti-SP9X) and for the whole cell extracts (blot anti-USP9X and beta-actin) are identical to the previously provided figure or might reflect a longer exposure of the previous figure (XIAP). S/he also notes that only the results for the anti-FLAG IP appear to be different. The reviewer also notes that the revised Fig.1F blot was probably not probed with V5 but rather with USP9X, which would explain the seemingly V5-reactive band in the mock lane.

2) We also noted a possible undeclared splicing (juxtaposition of blots not originally adjacent) in a figure panel.

Please provide a detailed explanation of the occurrences indicated in points 1 and 2 above, corrected image files and source data and where relevant the new experiments asked for by the reviewer.

I also ask you to please introduce the following amendments in your manuscript:

1) Please indicate wherefrom the magnification insets in Fig.3 D and E were derived in the original image.

2) As per our Author Guidelines, the description of all reported data that includes statistical testing must state the name of the statistical test used to generate error bars and P values, the number (n) of independent experiments underlying each data point (not replicate measures of one sample), and the actual P value for each test (not merely 'significant' or ' $P < 0.05$ ').

3) We encourage the publication of source data, particularly for electrophoretic gels and blots, with the aim of making primary data more accessible and transparent to the reader. To this effect, please provide a PDF file per figure that contains the original, uncropped and unprocessed scans of the gels used in the manuscript. The PDF files should be labeled with the appropriate figure/panel number, and should have molecular weight markers; further annotation may be useful but is not essential. The PDF files will be published online with the article as supplementary "Source Data" files. If you have any questions regarding this just contact me.

I look forward to reading a revised form of your manuscript as soon as possible.

***** Reviewer's comments *****

Referee #1 (Comments on Novelty/Model System):

I have some remaining technical issues (see below)

Referee #1 (Remarks):

The authors have addressed most of my concerns, but there still are some issues that need to be addressed.

Major issues :

1) The authors have revised Fig. 1D and now provide data which they claim are from a new experiment ('we ... repeated this experiment under conditions in which equal amounts of either mutant or wt forms of XIAP were precipitated and now clearly demonstrate that both XIAP mutants (...) do not interact with USP9X^a. I notice, however, that the blots for the IP (blot anti-SP9X) and for the whole cell extracts (WCE, blot anti-USP9X and beta-actin) are identical to the previously provided figure, or correspond to a stronger exposure of the previous figure (XIAP). Only the results for IP anti-FLAG have been exchanged. These are thus NOT data from a new experiment. Please show the complete dataset for a new experiment.

2) The new blot in revised Fig. 1F was probably not probed with V5 but rather with USP9X, which would explain the seemingly V5-reactive band in the mock lane. Please show the blot for V5, too.

3) The background for B220 staining looks very different in Fig. 3D and Figure 3E, I don't think we can conclude from these figures that B-cells are present in the samples.

Minor issues :

1) A minor issue to be fixed concerns aa numbering in Fig. EV1B. The authors refer in the text to aa 163 to 230, but the corresponding figure shows constructs with deletions of aa 152-323, which would encompass both the BIR2 and BIR3 domains and not only the BIR2 domain (as stated in the text).

Referee #2 (Remarks):

The authors have addressed the concerns raised and the new experiments support the conclusions of the study.

I am enclosing a revised version of our manuscript (Engel et al., EMM-2015-06047-V2) titled "USP9X stabilizes XIAP to regulate mitotic cell death and chemoresistance in aggressive B-cell lymphoma" in which we have addressed all remaining issues raised by the referees.

With regard to your point of undeclared splicing, we found this error in Fig. 1C (anti-Flag Blot in the WCE). A wrong panel was mistakenly inserted during the assembly process. I have attached a new version of this figure in which the correct anti-Flag Blot is shown. Importantly, this panel shows exactly the same results. In addition, I provide the uncropped and unprocessed scans of this figure.

Please find below our detailed point-by-point response to the reviewers' comments.

Reviewer #1:

The authors have addressed most of my concerns, but there still are some issues that need to be addressed.

Major issues :

"1) The authors have revised Fig. 1D and now provide data which they claim are from a new experiment (« we ... repeated this experiment under conditions in which equal amounts of either mutant or wt forms of XIAP were precipitated and now clearly demonstrate that both XIAP mutants (...) do not interact with USP9X ». I notice, however, that the blots for the IP (blot anti-SP9X) and for the whole cell extracts (WCE, blot anti-USP9X and beta-actin) are identical to the previously provided figure, or correspond to a stronger exposure of the previous figure (XIAP). Only the results for IP anti-FLAG have been exchanged. These are thus NOT data from a new experiment. Please show the complete dataset for a new experiment."

The indicated issue of Figure 1D is not a problem with the figure, but instead with our reply to this reviewers comment in the rebuttal letter. This figure indeed represents the original experiment demonstrated in our previous submission, but with longer exposures and larger panels of the FLAG-Blots to show that the XIAP mutants (XIAP-G188E and XIAP-G188R) are more ubiquitylated, as expected, as they can no longer be deubiquitylated by USP9X. Thus, the expression of the basal, non-ubiquitylated forms of these mutants is slightly lower than that of WT XIAP, explaining the somewhat reduced levels of both basal proteins in the figure of the previous submission. Together with the ubiquitylated species of the proteins, there is no difference in expression between the WT and mutant forms of XIAP. As a matter of fact, we indicate this point in the figure legend to Figure 1D. When writing the rebuttal letter, we mistakenly indicated that this was a new experiment, rather than explaining this context.

"2) The new blot in revised Fig. 1F was probably not probed with V5 but rather with USP9X, which would explain the seemingly V5-reactive band in the mock lane. Please show the blot for V5, too."

With regard to Figure 1F, the blot indicated to be blotted with anti-V5 was indeed blotted against with the anti-V5 antibody. This antibody gives an unspecific band at the size of USP9X. We have specified this context in the current submission.

3) The background for B220 staining looks very different in Fig. 3D and Figure 3E, I don't think we can conclude from these figures that B-cells are present in the samples.

We repeated the B220 stainings of Fig. 3D and 3E and now more clearly demonstrate B-cell origin of the respective tumor specimens.

Minor issues :

1) A minor issue to be fixed concerns aa numbering in Fig. EV1B. The authors refer in the text to aa 163 to 230, but the corresponding figure shows constructs with deletions of aa 152-323, which would encompass both the BIR2 and BIR3 domains and not only the BIR2 domain (as stated in the text).

We appreciate the reviewers comment on this incoherence. This is corrected in the current version of the manuscript and we state “Mapping studies using different deletion mutants narrowed the USP9X binding motif to the BIR2 and BIR3 domains of XIAP (aa152 - 323) (Fig. EV 1B).”.

Reviewer #2:

“The authors have addressed the concerns raised and the new experiments support the conclusions of the study.”

We are pleased to hear that this reviewer is now satisfied with the revised version of our manuscript.

I sincerely appreciate the constructive suggestions made by the reviewers and hope that our manuscript is now suitable for publication.

I am looking forward to hearing from you.

Acceptance

17 May 2016

Please find enclosed the final report on your manuscript. We are pleased to inform you that your manuscript is accepted for publication and will soon be sent to our publisher to be included in the next available issue of EMBO Molecular Medicine.

We would like to remind you that as part of the EMBO Publications transparent editorial process initiative, EMBO Molecular Medicine will publish a Review Process File online to accompany accepted manuscripts.

=> Please check and confirm as soon as possible whether or not we can publish the figure(s) you included in your point-by-point response(s) as part of this file or if you want to exclude it/them. Also, in case that you may not want that Review Process file to be published at all, please immediately inform us via e-mail.

If you want to receive an e-mail alert regarding its publication as well as other EMBO Mol Med content, register here: <http://embomolmed.embopress.org/alerts>

Congratulations on your interesting work.

***** Reviewer's comments *****

Referee #1 (Remarks):

The authors have now adequately adressed all of my concerns

Corresponding Author Name: Florian Bassermann

Manuscript Number: EMM-2015-06047-V2